# Honokiol Inhibits Colorectal Cancer Cell Growth: Involvement of Hsp27 as a Molecular Target

**DOI:** 10.3390/cimb47110921

**Published:** 2025-11-05

**Authors:** Youngbin Kim, Hyeon Du Jang, Da Hyeon An, Hyun Seo Lee, Hong-Gyum Kim, Sun Eun Choi

**Affiliations:** 1Department of Forest Biomaterials Engineering, Kangwon National University, Chuncheon 24341, Gangwon-do, Republic of Korea; ybkim_@shebah.co.kr (Y.K.); wkdgusen98@naver.com (H.D.J.); 202416560@kangwon.ac.kr (D.H.A.); hyunseo2002@kangwon.ac.kr (H.S.L.); 2SHEBAH BIOTECH Inc., 50-1, Geodudanji 1-gil, Dongnae-myeon, Chuncheon 24398, Gangwon-do, Republic of Korea; 3NovelVita Inc., #310, Bio-2-dong, 32, Soyanggang-ro, Chuncheon 24232, Gangwon-do, Republic of Korea; 4Dr.Oregonin Inc., #802 Bodeum Hall, Kangwondaehakgil 1, Chuncheon 24341, Gangwon-do, Republic of Korea

**Keywords:** honokiol, Hsp27, apoptosis, CRC cells, anti-cancer effect

## Abstract

Background/Objectives: Honokiol (HK), a bioactive phenolic compound, exhibits significant anti-cancer properties. This study aimed to investigate the anti-cancer effects of HK in colorectal cancer (CRC) cells by focusing on its direct interaction with heat shock protein 27 (Hsp27) as a molecular target, and to elucidate the underlying mechanisms involved. Methods: HK was isolated via silica/ODS chromatography. Anchorage-independent growth of CRC cells was quantified using a soft agar assay with increasing HK concentrations. Apoptosis and cell cycle were analyzed by flow cytometry, and cell viability by MTS assay. Hsp27 binding to HK was validated by pull-down assay with HK-conjugated Sepharose 4B beads. Hsp27 knockdown was performed using lentiviral shRNA in CRC cells. Molecular docking of HK-Hsp27 interaction employed Schrödinger Suite 2016. Protein expressions, including chaperone and apoptotic proteins, were evaluated by Western blotting. Results: HK dose-dependently suppressed anchorage-independent growth of CRC cells and induced G_0_/G_1_ arrest. It triggered apoptosis through cytochrome c release, PARP cleavage, and Bcl-2 downregulation. HK directly bound to the α-crystallin domain of Hsp27 at Asn102 and His103 residues, confirmed by computational molecular docking and site-directed mutagenesis. Hsp27 knockdown in CRC cells dramatically reduced anchorage-independent growth. HK markedly decreased Hsp27 protein levels while having less effect on other heat shock proteins in CRC cells. Conclusions: HK exerts anti-cancer effects in CRC cells, associated with Hsp27 inhibition, resulting in suppressed cell growth and increased apoptosis. This interaction between HK and Hsp27 may support a mechanistic foundation supporting the potential utility of HK as a natural therapeutic agent for CRC.

## 1. Introduction

Colorectal cancer (CRC) is the third most commonly diagnosed malignancy and the second leading cause of cancer-related mortality worldwide, with an estimated 1.93 million new cases in 2020, accounting for approximately 10% of all cancer diagnoses globally [1]. CRC comprises a multi-step process that includes genetic alterations and genetic instability that occur during its development [2]. It is also associated with various environmental factors, such as diet and personal habits. Molecular studies focusing on CRC have shown that multiple genetic alterations, including changes in β-catenin, adenomatous polyposis coli protein (APC), and *c-myc* are involved in colonic tumorigenesis [3]. CRC remains difficult to manage due to high rates of recurrence, metastasis, and drug resistance. Furthermore, the side effects of chemotherapy limit its effectiveness, necessitating the development of new therapeutic approaches.

Given these limitations, natural compounds have attracted attention as safer, multi-targeted therapeutic options for CRC. Several phytochemicals, such as piceatannol and curcumin, have demonstrated tumor-suppressive and anti-inflammatory activities in preclinical CRC models [4,5]. Among them, honokiol (HK), a biphenolic compound isolated from *Magnolia officinalis*, has drawn particular interest due to its low toxicity and potential suitability for clinical applications [6]. It inhibits tumor growth and metastasis in CRC animal models by inducing apoptosis and modulating autophagy pathways [7]. HK has been shown to possess potent anti-cancer, anti-proliferative, anti-inflammatory, and pro-apoptotic properties in a wide range of human cancer cells [8]. These findings underscore the promise of natural compounds as complementary or alternative therapies for CRC.

In CRC models, HK has demonstrated significant anti-cancer efficacy through multiple mechanisms. At the cellular level, its anti-proliferative effect is mediated by BMP7 upregulation via the TGF-β1/p53 signaling pathway [9], and it also triggers ferroptosis by suppressing Glutathione Peroxidase 4 (GPX4) activity [10]. Moreover, HK enhances cetuximab sensitivity in KRAS-mutant CRC preclinical model by destabilizing the SNX3 retromer complex [7], and semisynthetic honokiol thioethers have shown potent activity against HCT116 CRC cells through inhibition of YAP transcription and expression [11]. Collectively, these findings underscore the diverse mechanisms by which HK suppresses CRC progression. Nonetheless, the direct molecular targets and precise mechanisms underlying its anti-cancer actions remain to be fully defined.

Heat shock protein 27 (Hsp27), encoded by the *HSPB1* gene, has demonstrated anti-aggregation properties as a molecular chaperone for damaged proteins [12]. It plays a role in anti-apoptotic functions by suppressing the activity of apoptotic proteins, including caspases [13]. Overexpression of Hsp27 inhibits cytochrome c release from mitochondria, resulting in protection from apoptosis [14]. Hsp27 is overexpressed in several cancer types, including colon, prostate, pancreatic, and lung cancer, as well as various other solid tumors, particularly those resistant to chemotherapy [15]. Its overexpression is associated with tumorigenesis, metastasis, multi-drug resistance, cytoprotection, and inhibition of apoptosis. These effects lead to poor prognosis and therapeutic resistance in these cancers.

Hsp27 expression is markedly elevated in patients with ovarian cancer compared to healthy individuals, with levels positively correlating with tumor progression [16]. Marked Hsp27 overexpression in CRC supports its potential as a biomarker for diagnosis and targeted therapy, as revealed by immunohistochemistry [17]. Similarly, in breast cancer, Hsp27 is upregulated in tumor tissues, linked to poor prognosis, and has been shown to promote tumor growth and metastasis both in vitro and in vivo [18].

Conversely, suppression of Hsp27 by inhibitors effectively reduced the progression of prostate and pancreatic carcinoma cells [19]. Treatment of human prostate cancer cells, which highly express Hsp27, with atractylenolide I has been reported to suppress Hsp27 expression, thereby reducing cell viability and inducing apoptosis [20]. Furthermore, Hsp27 silencing increased the sensitivity of oncogenic-activated carcinoma cells to specific drug treatments [21]. In CRC, downregulation of Hsp27 significantly promoted apoptosis, suppressed tumor growth, and increased the sensitivity of cancer cells to chemotherapeutic agents such as 5-fluorouracil (5-FU) and vincristine [22].

HK is known for its anti-cancer effects in CRC cells, but the molecular targets and interactions remain unclear. Based on recent reports highlighting the critical role of Hsp27 in CRC progression and therapy resistance, we hypothesize that HK exerts anti-cancer effects potentially through interaction of Hsp27. This study focuses on investigating the interaction between HK and Hsp27 to elucidate the molecular mechanism through which HK may modulate the function of Hsp27 in CRC.

## 2. Materials and Methods

### 2.1. Isolation and Purification of HK

The barks of *M. officinalis* were sourced from Seoul Yakryeong market and certified by Professor Choi (Department of Forest Biomaterials Engineering, Kangwon National University). The isolation and purification of HK from *M. officinalis* were performed with reference to the method described [23], with slight modifications. Briefly, *M. officinalis* bark was extracted using 60% EtOH. The extract was concentrated under reduced pressure using a rotary evaporator and then freeze-dried (Ilshin Biobased Co., Ltd., Dongducheon-si, Gyeonggi-do, Republic of Korea).

For solvent fractionation, the dried EtOH extract was dissolved in distilled water and mixed with ethyl acetate at a 1:1 (*v*/*v*) ratio. The ethyl acetate (EA) layer was collected and used as the starting material for HK purification. Purification was performed in two steps using medium-performance liquid chromatography (MPLC) with a Yamazen Smart Flash AKROS MPLC system (Itachibori, Nishi-ku, Osaka, Japan). In the first step, a silica gel column was used as the stationary phase, and the mobile phase was chloroform/methanol (94:6, *v*/*v*) at a flow rate of 10 mL/min for 20 min.

The injection volume was 1 mL, and detection was carried out at 254 nm. In the second step, an octadecylsilyl (ODS) column was used as the stationary phase. The mobile phases were distilled water (A) and methanol (B), with the following gradient: 0–10 min, 70% B; 10–50 min, 100% B; 50–60 min, 100% B; 60–62 min, 70% B; 62–70 min, 70% B. The flow rate was maintained at 10 mL/min, with an injection volume of 0.5 mL and detection at 254 nm.

### 2.2. Analysis of HK

#### 2.2.1. Qualitative Analysis on Thin Layer Chromatography (TLC)

Thin-layer chromatography (TLC) was performed to qualitatively analyze both the *M. officinalis* 60% EtOH extract and the isolated compound. Each sample (3.5 mg) was dissolved in 1 mL of methanol to prepare 3500 ppm solutions. The solutions were spotted onto silica gel plates and developed using benzene/methanol (9:1, *v*/*v*) as the mobile phase. After development and complete drying, the plates were visualized under a UV lamp at 254 nm. For further characterization, three different color reagents—10% H_2_SO_4_, *ρ*-anisaldehyde-H_2_SO_4_, and FeCl_3_-were applied to the plates.

#### 2.2.2. LC-MS/MS Analysis

LC-MS/MS analysis of the *M. officinalis* 60% EtOH extract and the isolated compound was performed using a QTRAP 4500 system (AB SCIEX, Billerica, MA, USA) equipped with a HECTOR C18 analytical column (5 µm particle size) and a Phenomenex KJ0-4282 guard column, as described [23]. The mobile phases consisted of 1% formic acid in distilled water (solvent A) and acetonitrile (solvent B). The flow rate was set to 1.0 mL/min, and the total run time was 40 min. The gradient program was as follows: 0–0 min, 10% B; 0–5 min, 40% B; 5–20 min, 75% B; 20–25 min, 80% B; 25–30 min, 10% B; and 30–40 min, 10% B.

#### 2.2.3. NMR Analysis

Nuclear magnetic resonance (NMR) spectroscopy was conducted using a 400 MHz FT-NMR spectrometer (JNM-ECZ400S/L1, JEOL Ltd., Tokyo, Japan). The spectra were recorded at 400 MHz for ^1^H and 100 MHz for ^13^C. Deuterated chloroform (CDCl_3_) was used as the solvent.

#### 2.2.4. Quantitative Analysis on High-Performance Liquid Chromatography (HPLC)

HPLC was performed using a Waters 1525 system equipped with a Waters 996 photodiode array (PDA) detector (Waters, Milford, MA, USA). Chromatographic separation was achieved on a Hector C18 column (5 µm particle size) fitted with a KJ0-4282 guard column (Phenomenex, Torrance, CA, USA). The mobile phase consisted of solvent A (distilled water containing 1% formic acid) and solvent B (acetonitrile). Gradient elution was carried out at a flow rate of 1.0 mL/min under the following program: 0 min, 10% B; 0–5 min, 40% B; 5–20 min, 75% B; 20–25 min, 80% B; 25–30 min, 10% B; and 30–40 min, 10% B. The injection volume was set at 10 µL, and detection was monitored at 254 nm.

For sample preparation, the extract, isolated compound, and HK standard were dissolved in methanol (1 mg/mL). Serial dilutions of the HK standard were prepared to yield concentrations of 1 to 100 ppm for calibration curve construction.

### 2.3. Reagents and Antibodies

Pre-stained protein markers, 10× Trypsin-EDTA, 100× penicillin/streptomycin and protease inhibitor cocktails were from GenDEPOT (Houston, TX, USA). Basal Medium Eagle (BME), RPMI1640 and McCoy’s 5A medium purchased at Sigma Aldrich (St. Louis, MO, USA). Fetal bovine serum (FBS) and polyvinylidene difluoride (PVDF) membrane were from Atlanta Biologicals (Lawrenceville, GA, USA) and Millipore (Bedford, MA, USA), respectively. Primary antibodies for detecting Hsp27 (#sc-13132), Hsp70 (#sc-24), β-actin (#sc-47778), mouse anti-rabbit IgG-HRP (#sc-2357), and rabbit anti-mouse IgG-HRP (#sc-358914) were purchased from Santa Cruz Biotechnology (Santa Cruz, CA, USA). The antibodies against PARP (#9542), cytochrome C (#4272), cyclin D (#2922) and Bcl-2 (#2876) were from Cell Signaling (Beverly, MA, USA). Anti-GRP78 BiP antibody was purchased from Abcam (#ab21685, Cambridge, UK). CNBr-Activated Sepharose 4B beads were purchased from GE Healthcare Life Sciences (#17-0430-01, Marlborough, MA, USA). The 6x-His Tag Monoclonal Antibody (HRP-conjugated His-HRP, #MA1-21315), Xpress Monoclonal Antibody (#R910-25), BCA protein assay kit (#23227), and Lipofectamine (#18324012) were obtained from Thermo Fisher Scientific (Waltham, MA, USA). QuickChange II Site-directed Mutagenesis Kit was purchased Agilent Technologies (#200523, Santa Clare, CA, USA).

### 2.4. Cell Lines and Cell Culture

CRC cell lines HCT-15 (ATCC CCL-225), DLD-1 (ATCC CCL-221), HCT116 (ATCC CCL-247), and HT-29 (ATCC HTB38) were purchased from the American Type Culture Collection (ATCC; Manassas, VA, USA). HCT-15 and DLD-1 cells were cultured in RPMI 1640 medium supplemented with 10% FBS and 100 units/mL penicillin–streptomycin. HCT116 and HT-29 cells were cultured in McCoy’s 5A medium supplemented with 10% FBS and 100 units/mL penicillin/streptomycin. HEK293T (ATCC CRL-3216) cells were cultured in DMEM supplemented with 10% FBS and 100 units/mL penicillin–streptomycin. All cells were maintained at 37 °C in a humidified incubator with 5% CO_2_.

### 2.5. Construction of Hsp27 Stable Knockdown CRC Cells

The *pLKO.1* lentiviral vector for Hsp27 knockdown was purchased from Addgene (#8453, Watertown, MA, USA). To knock down Hsp27 in HCT-15 or DLD-1 CRC cells, 1 μg of *shRNA-HSP27* (*sh-Hsp27*) or a non-targeting *shRNA* control (*sh-NT*) vector and lentiviral packaging plasmids (0.8 μg of *psPAX2* and 0.2 μg of *PMD2.G*) were mixed with 5 μL of Lipofectamine then transfected into HEK293T cells. The full hairpin sequence of the *sh-NT* and *sh-Hsp27* are shown in Table 1.

Cells were incubated overnight at 37 °C in a 5% CO_2_ incubator. Viral supernatants were collected at 24 and 48 h after refreshing the medium. CRC cells were infected with Hsp27 knockdown viral particles in the presence of 8 μg/mL polybrene (#S2667, Merck, Darmstadt, Germany). Infected clones were selected in medium containing 2 μg/mL puromycin for 10 days, with the medium refreshed every two days to maintain puromycin. The expression level of Hsp27 was confirmed by Western blot analysis.

### 2.6. Anchorage-Independent CRC Cells Growing Assay

CRC cells or Hsp27-knockdown CRC cells (8 × 10^3^/mL) were cultured in 1 mL of 0.3% (top layer) and 0.5% (base layer) Basal Medium Eagle (BME) agar supplemented with 10% FBS and treated with HK at final concentrations of 0, 10, 20, 30, 40, or 50 μM. The treatment concentration of HK was determined based on previous research findings [24]. The cultures were incubated at 37 °C in a humidified atmosphere with 5% CO_2_ for 7 days, after which cell colonies were examined under a microscope and quantified using the Image-Pro PLUS software (v.6, Media Cybernetics, Rockville, MD, USA).

### 2.7. MTS Assay

CRC cells (1 × 10^3^) were seeded into 96-well plates in 100 μL of McCoy’s 5A or RPMI-1640 medium supplemented with 10% FBS and incubated overnight at 37 °C in a humidified atmosphere containing 5% CO_2_. After overnight incubation, cells were treated with HK at concentrations of 0, 10, 20, 30, 40, or 50 µM and cultured for 48 or 72 h. At the indicated time point, 20 µL of the CellTiter 96^®^ Aqueous One Solution Reagent (#G3582, Promega, Madison, WI, USA) was added to each well, followed by incubation for 1 h under the same conditions. The reaction was terminated by adding 25 µL of 10% SDS solution, and absorbance was measured at 492 nm with a reference wavelength of 690 nm.

### 2.8. Western Blot Analysis

Each cell was washed with PBS buffer and lysed with RIPA buffer containing protease inhibitor cocktail on ice. The lysate recovered by centrifugation, and protein concentration was measured using BCA protein assay kit. Equal amounts of protein were resolved by SDS-PAGE and transferred onto PVDF membranes. Membranes were first washed with TBS-T buffer (20 mM Tris, 137 mM NaCl, pH 7.6, containing 0.1% Tween 20) and then incubated in blocking buffer for 1 h. The membranes were subsequently incubated overnight at 4 °C with the appropriate primary antibodies diluted 1:1000 in TBS-T containing 5% BSA to detect chaperone proteins, including Hsp27, cytochrome c, Bcl-2, cleaved PARP, anti-His HRP, Xpress monoclonal antibody, GRP78, or β-actin. After primary antibody incubation, membranes were washed with TBS-T and incubated with the secondary antibodies diluted 1:5000 at room temperature for 1 h. The Western blots were visualized using an Amersham ECL Prime Western Blotting Detection Reagent (#RPN2232, Cytiva, Marlborough, MA, USA) following hybridization with the appropriate horseradish peroxidase-conjugated secondary antibody.

### 2.9. Ex Vivo Pull-Down Assay

Each cell lysate (500 μg) was gently rocked and incubated with HK-conjugated Sepharose 4B or vehicle-conjugated Sepharose 4B beads (as control) in non-denaturing lysis buffer (20 mM Tris-Cl pH 8.0, 137 mM NaCl, 1% NP-40 and 2 mM EDTA including protease inhibitors) for overnight at 4 °C. The beads were washed 4 times with non-denaturing lysis buffer, and the binding was visualized by Western blotting.

### 2.10. Site-Directed Mutagenesis

Single or double alanine substitutions at Asn102 and His103 in *pcDNA4C-Hsp27* were introduced using specific point-mutated primers with the QuickChange II Site-Directed Mutagenesis Kit. The sequences of the primers are shown in Table 2.

PCR reactions were performed in a 50 μL volume containing 50 ng of *pcDNA4c-Hsp27* plasmid, 125 ng of each primer, 1 μL of dNTP mix, 1 μL of QuickSolution, 1 μL of *PfuUltra* High-Fidelity DNA polymerase, and 5 μL of 10× Reaction Buffer. Thermal cycling was carried out under the following conditions: initial denaturation at 95 °C for 30 s; 18 cycles of 95 °C for 30 s, 55 °C for 1 min, and 68 °C for 6 min; followed by a final extension at 68 °C for 5 min.

After amplification, 1 μL of DpnI was added directly to the PCR mixture and incubated at 37 °C for 1 h to digest the *pcDNA4c-Hsp27* plasmid. The DpnI-treated solution (2 μL) was transformed into competent *E. coli* cells, which were plated on LB agar containing ampicillin (100 μg/mL) and incubated overnight at 37 °C. Plasmid DNA was purified from individual colonies using a QIAprep Spin Miniprep Kit (#27106, Qiagen, Hilden, Germany) according to the manufacturer’s instructions. All mutated constructs were confirmed by DNA sequencing analysis (Genewiz, South Plainfield, NJ, USA).

### 2.11. Cell Cycle Analysis

CRC cells (1 × 10^5^/mL) were seeded in 60 mm dishes and cultured overnight at 37 °C in a 5% CO_2_ incubator. Cells were treated with or without 30 μM HK for 48 h. Based on preliminary experimental results, treatment with 30 μM HK for 48 h was confirmed as the optimal condition to achieve the greatest impact on the cell cycle. After treatment, cells were harvested using trypsin, fixed with ice-cold methanol, stained with propidium iodide (PI), and analyzed for cell cycle distribution by flow cytometry using a FACSymphony S6 system (Becton Dickinson, Franklin Lakes, NJ, USA) and FACSDiva Software (version 8.0).

### 2.12. Apoptosis Analysis

HCT-15 and DLD-1 cells (1 × 10^5^/mL) were seeded in 60 mm dishes and cultured at 37 °C in a 5% CO_2_ incubator for 48 h. Cells were cultured in serum-free medium for 12 h followed by treatment with HK at 30 or 50 μM in serum-free medium for 24 h. After treatment, the conditioned medium was collected, and attached cells were harvested using 0.025% trypsin with 5 mM EDTA in phosphate-buffered saline (PBS). Trypsinization was stopped by adding 2 mL of 5% FBS in PBS. Cells were washed by centrifugation at 1000 rpm for 5 min and processed for detection of early and late apoptosis using annexin V-fluorescein isothiocyanate (FITC) and PI staining and then analyzed by flow cytometry as described above.

### 2.13. Computational Modeling of Hsp27 with HK

First the three-dimensional (3-D) structure of Hsp27 (PDB ID:4MJH) was downloaded from the Protein Data Bank [25]. The structure was an X-ray crystal structure with a resolution of 2.6 Å of the human Hsp27 core domain in complex with the C-terminal peptide [26]. This raw PDB format structure was converted into an all-atom, fully prepared receptor model structure for docking using the Protein Preparation Wizard in Schrödinger Suite 2016 [27]. Hydrogen atoms were added, consistent with a pH of 7 and all water molecules were removed. The SiteMap program suggested a binding pocket of Hsp27 that was generated for docking. The HK compound was prepared for docking by default parameters using the LigPrep program. Then, the docking of HK with Hsp27 was accomplished with default parameters under the extra precision (XP) mode using the program Glide, respectively. Herein, we could get the best-docked representative structures.

### 2.14. Statistical Analysis

Each in vitro experiment was performed in triplicate. Quantitative data are presented as mean ± S.E.M. or mean ± S.D., unless otherwise noted. Statistical analyses were conducted using one-way analysis of variance (ANOVA) and Student’s *t*-test, respectively. Statistical significance between groups was determined by a non-parametric test where appropriate. The number of replicates for each sample is indicated in the figure legends. A *p*-value of <0.05 was considered statistically significant.

## 3. Results

### 3.1. Isolation and Structural Characterization of HK

#### 3.1.1. Purification of HK

A total of 257.75 g of extract was obtained from 3 kg of *M*. *officinalis* bark using 60% ethanol (EtOH) extraction. Of this, 150 g was used for the isolation and purification of HK, and solvent fractionation was performed using ethyl acetate. A two-step purification process involving MPLC was conducted (Appendix A), and the final four fractions were obtained using an ODS column. The yields were as follows: ODS Fr.1, 62 mg (yield: 1.95%); ODS Fr.2, 84.6 mg (yield: 2.67%); ODS Fr.3, 51.3 mg (yield: 1.62%); and ODS Fr.4, 85.7 mg (yield: 2.70%).

#### 3.1.2. Qualitative Analysis of Isolated Compound

Thin-Layer Chromatography (TLC) analysis showed the isolated compound and the extract shared the same Rf value, with the isolated compound presenting as a single pure spot (Appendix A).

LC-MS/MS analysis was conducted exclusively on ODS Fr.1 in positive ion mode. The analysis revealed a molecular ion peak at 267.4 *m*/*z*, which is within the acceptable error range for the standard molecular mass of HK (Figure 1). This result confirms the successful isolation and identification of HK from *M. officinalis* bark extract using LC-MS/MS analysis.

#### 3.1.3. NMR Analysis and Structural Elucidation of HK

Structural elucidation of ODS Fr.1 was performed using both ^1^H-NMR and ^13^C-NMR spectroscopy, which identified the compound as HK. These results confirm that the major constituent of ODS Fr.1 is HK.

**Honokiol** ^1^H NMR (400 MHz, CDCl_3_) δ 7.22 (d, J = 8.2 Hz, 2H, H-2′/6′), 7.07–7.00 (m, 2H, H-3′/5′), 6.90 (dd, J = 8.0, 6.2 Hz, 2H, H-3/5/6), 6.00 (dddt, J = 27.8, 16.9, 10.1, 6.6 Hz, 2H, H-8), 5.25–5.02 (m, 6H, H-9a/b, H-9′a/b), 3.46 (d, J = 6.4 Hz, 2H, H-7), 3.34 (d, J = 6.8 Hz, 2H, H-7′). & ^13^C NMR (100 MHz, CDCl_3_) δ 153.99 (C-4′), 150.81 (C-4), 137.80 (C-1), 136.00 (C-1′), 132.21 (C-3), 131.14 (C-5), 130.22 (C-6), 129.61 (C-2′), 128.83 (C-3′), 128.57 (C-5′), 127.71 (C-6′), 126.36 (C-2), 116.93 (C-3a), 116.59 (C-3b), 115.59 (C-5a), 115.55 (C-5b), 39.42 (C-7), 35.18 (C-7′).

#### 3.1.4. Quantitative Chromatographic Analysis of *M. officinalis* 60% EtOH Extract (HPLC)

The HK standard used for calibration was isolated by preparative purification and confirmed to have high purity (approximately 95%) by HPLC analysis. Using this standard, a calibration curve was established by plotting peak areas against concentration at six different levels, and the relationship was determined by the least-squares method. The resulting calibration equation was Y = 23,498X − 10,169, with an R^2^ value of 0.9998 (Appendix A) indicating excellent linearity. Chromatograms for each concentration level are shown in Appendix A, and all relevant data are provided in Appendix A. Based on this calibration, the HK content in the 60% EtOH extract of *M. officinalis* was determined to be 12.02 ppm (Figure 2).

### 3.2. Biological Effects of HK on CRC Cells

#### 3.2.1. HK Suppresses the Growth of CRC Cells

We performed an anchorage-independent cell growth assay to determine whether HK could inhibit the growth of CRC cells in soft agar. A previous study demonstrated that non-tumorigenic human colonic epithelial cells (HCoEpiCs) treated with HK at concentrations up to 50 μM exhibited no cytotoxicity [28]. Based on these findings, we subsequently evaluated the growth inhibitory effects of HK on CRC cells. The results demonstrated that HK markedly inhibited both the number and size of colonies formed by HCT-15, DLD-1, HT-29, and HCT116 cell lines (Figure 3a), indicating that HK dose-dependently suppresses the anchorage-independent growth, which reflects the tumorigenic potential of CRC cells.

To assess the effect of HK on cell viability and cell cycle progression in CRC cells, we analyzed changes in cell growth and cell cycle distribution following HK treatment. The results indicated that HK inhibited CRC cell growth in a dose-dependent manner (Figure 3b). Furthermore, HK induced G0/G1 phase arrest in CRC cells (Figure 3c). These findings suggest that HK is a potential compound for effectively suppressing CRC cell proliferation, prompting us to further investigate the molecular mechanisms underlying the activity of HK.

#### 3.2.2. HK Induces Apoptosis in CRC Cells

We next investigated whether HK induces apoptosis in HCT-15 and DLD-1 CRC cells. These cell lines were selected for apoptosis because they display higher sensitivity to apoptosis induction and possess well-characterized apoptotic pathways compared with HCT116 and HT-29 cells [29]. The results showed that HK significantly increased apoptosis in CRC cell lines, including HCT-15 and DLD-1 cells (Figure 4a). As shown in Figure 4b, HK treatment markedly elevated the protein levels of cytochrome c and cleaved PARP, indicating that HK induces substantial apoptosis through pathways involving cytochrome c release and PARP cleavage. Conversely, the levels of the anti-apoptotic protein Bcl-2 were reduced in HCT-15 and DLD-1 cells following HK treatment (Figure 4b). Taken together, these findings suggest that HK induces apoptosis and thereby contributes to the elimination of CRC cells.

### 3.3. Molecular Interaction Between HK and Hsp27

#### 3.3.1. Identification for HK Direct Binding Protein

We investigated the direct protein targets of HK in CRC cells. Lysates from HCT116 cells were incubated with HK-conjugated Sepharose 4B beads or control Sepharose beads. Proteins bound to HK were pulled down and subsequently identified by LC-MS/MS analysis. The results demonstrated that HK directly binds to heat shock proteins (HSPs), including Hsp27 and Hsp70 (Table 3).

#### 3.3.2. HK Suppresses Hsp27 Protein Expression Level

To assess the effect of HK on the HSPs expression in CRC cells, we measured the protein levels of key HSPs, including Hsp27, Hsp70 and GRP78, following HK treatment. Interestingly, HK significantly reduced Hsp27 protein levels across CRC cells, whereas Hsp70 expression was modestly decreased only in DLD-1 cells. In contrast, GRP78 protein levels remained unaffected by HK treatment (Figure 5). These results suggest that HK specifically downregulates Hsp27 protein levels, potentially by regulating its transcription or protein stability. Previous studies have shown that HK directly binds GRP78, an endoplasmic reticulum chaperone, modulating its activity and triggering ER stress-mediated apoptosis [30]. Collectively, these findings highlight the selective regulation of HSPs by HK in CRC cells, with Hsp27 serving as a primary target over other HSPs.

#### 3.3.3. HK Directly Binds to Hsp27

Based on the results shown in Figure 5, we hypothesized that the interaction between HK and Hsp27 plays a critical role in CRC cells and therefore, investigated the direct molecular interaction between HK and Hsp27. To clearly demonstrate the direct binding of HK to Hsp27, we performed an ex vivo pull-down binding assay. This assay was conducted using HEK293T cells with transient overexpression of Hsp27 and CRC cells expressing endogenous levels of Hsp27.

First, HEK293T cells were transfected with the *pcDNA4C-Hsp27* plasmid to overexpress Hsp27. Cell lysates were incubated with HK-conjugated Sepharose 4B beads or control Sepharose 4B beads to assess direct binding. HK-conjugated beads specifically pulled down the overexpressed Hsp27, whereas control beads did not. This result confirms a direct and specific interaction between HK and Hsp27 (Figure 6a).

Second, to validate the physiological relevance of this interaction, we examined endogenous Hsp27 binding in CRC cell lines, including HCT-15 and DLD-1. These cells showed markedly higher expression of Hsp27 compared to normal colon fibroblast CCD-18Co cells (Appendix A). Using lysates from these CRC cells, pull-down assays with HK-conjugated beads demonstrated specific binding of HK to endogenous Hsp27 protein, but control beads did not pull down Hsp27 (Figure 6b). This further confirms the specificity of the interaction under physiological expression conditions.

Together, these results validate that HK directly binds to Hsp27 both in an overexpressed system and at endogenous levels in CRC cells. This supports the specificity and biological relevance of the HK-Hsp27 interaction.

Next, to further elucidate how HK interacts with Hsp27, we performed molecular docking using the binding pocket of Hsp27. This approach followed several protocols included in Schrödinger Suite 2016. Based on the computational docking results, HK was found to form hydrogen bonds with Hsp27, indicating that HK may serve as a potential inhibitor of Hsp27. The docking images were generated using the UCSF Chimera program [31]. Simulation results revealed that the hydroxyl groups of HK interact with the core region of Hsp27, which corresponds to the central α-crystallin domain of Hsp27 rather than its C-terminal region. Specifically, HK binds to key residues, including Asn102 and His103, in this domain (Figure 7a).

To confirm these findings, we constructed single- and double-point mutants of Hsp27 at the Asn102 and His103 residues. Binding assays were performed to assess whether HK binds to both wild-type and mutant Hsp27. As shown in Figure 7b, the double mutant Hsp27 exhibited reduced binding to HK compared to wild-type Hsp27, indicating that both the Asn102 and His103 residues contribute significantly to HK binding.

Collectively, these results confirm that HK directly binds to the core region of Hsp27, with the Asn102 and His103 residues playing critical roles in mediating this interaction. The significant reduction in binding affinity observed in the double mutant highlights the specificity of HK for these key amino acids. These findings suggest that HK may modulate the function of Hsp27 by targeting its core binding site, potentially affecting its chaperone activity, stability, or role in oncogenic processes; however, further studies are needed to clarify the extent and biological implications of this modulation.

#### 3.3.4. Hsp27 Knockdown Suppresses CRC Cell Growth

Previous studies have demonstrated that Hsp27 protein is highly expressed in CRC cells [32], a finding we also confirmed in our own experiments (Appendix A). To investigate the role of Hsp27 in CRC cell growth, we established Hsp27 knockdown DLD-1 and HCT-15 cell lines using *sh-Hsp27* and compared them to cells infected with a control *sh-NT* Lenti-viral vector. The complete hairpin DNA sequences for both the non-target control (*sh-NT*) and the Hsp27 knockdown (*sh-Hsp27*) constructs are shown in Table 2. Results showed endogenous Hsp27 protein expression was reduced by approximately 90% in Hsp27 knockdown cells relative to control cells (Figure 8a). To assess the functional consequences of Hsp27 knockdown, we measured cell proliferation using the MTS assay. The results revealed that suppression of Hsp27 in DLD-1 and HCT-15 cells led to a markedly reduced proliferation rate compared to control cells (Figure 8b). Additionally, Hsp27 knockdown significantly inhibited anchorage-independent growth in soft agar compared to *sh-NT* controls (Figure 8c).

Overall, these data show that silencing Hsp27 impairs both the growth and anchorage-independent colony formation of CRC cells, underscoring the pivotal role of Hsp27 in CRC cell proliferation and survival.

## 4. Discussion

Various methods for the isolation and purification of HK have been reported in previous studies. In the present study, HK was isolated from *M*. *officinalis* bark by extraction with 60% EtOH, followed by solvent partitioning with ethyl acetate. Subsequently, a two-step purification process was performed using MPLC. Notably, instead of the conventional hexane mobile phase commonly used in previous studies, a chloroform/methanol mixture (94:6, *v*/*v*) was employed as the mobile phase. EtOH extraction of *M. officinalis* bark yielded 257.75 g of crude extract, from which 165.7 mg of HK was ultimately isolated, corresponding to a yield of 1.89 mg/g. This extraction yield is higher than the previously reported commercial product yield of 0.64 mg/g [33], suggesting the reliability and superiority of the isolation method used in this study. This approach therefore provides a potential alternative for efficiently obtaining HK with higher purity at the laboratory scale.

Previous investigations showed that HK exerted anti-cancer effects in a variety of different cancer types [8,34]. It reduced xenograft gastric tumor growth in mice by stimulating calpain-mediated glucose-regulated protein-94 (GRP94) cleavage [35]. HK also inhibited tumor growth by inducing apoptosis and suppressing HIF-1α-mediated glycolysis in human breast cancer cells [36]. In addition, HK suppressed the migration and invasion of H1299 lung cancer cells by disrupting the expression of matrix metalloproteinase 9 (MMP9) through HDAC6 modulation [37]. HK inhibited lung cancer growth by suppressing tumor cell PD-L1 expression, blocking the PD-1/PD-L1 pathway, and thereby enhancing anti-tumor immunity [38]. It has been shown to suppress the proliferation of CRC cells by inhibiting anoctamin 1/TMEM16A Ca^2+^-activated Cl^−^ channels [39]. Its anti-proliferative activity in SW620 cells is mediated by upregulation of BMP7 expression via the TGF-β1/p53 signaling pathway [22]. In this study, we demonstrated that HK significantly suppressed colony formation and cell growth in various CRC cell types (Figure 3).

HK has also been shown to induce apoptosis in different human cancer cells. It promoted PARP cleavage and caspase-3 activation in CRC cells [40]. It also induced apoptosis in the CT26 murine colon carcinoma cell line, and combined with cisplatin, it synergistically suppressed tumor progression in a CT26 mouse xenograft model [41]. In estrogen receptor (ER)-negative MDA-MB-231 breast cancer cells, HK induced apoptosis by inhibiting Bcl-2 [42]. Consistent with these findings, our study demonstrated that HK strongly induced apoptosis in CRC cells, leading to upregulation of pro-apoptotic molecules (cleaved PARP and cytochrome c) and downregulation of Bcl-2 (Figure 4).

Recent research has established that Hsp27 is upregulated in various cancer types, and this increase is frequently associated with the development of drug resistance. In breast cancer, Hsp27 expression is significantly higher in angiogenic cells compared to non-angiogenic counterparts. This observation suggests a functional role for Hsp27, as forced overexpression in non-angiogenic breast cancer cells has been shown to promote expansive tumor growth in vivo [43]. The meta-analysis showed that positive Hsp27 expression was significantly associated with the incidence of hepatocellular carcinoma (HCC), tumor differentiation, and α-fetoprotein levels in patients with HCC [44]. Immunohistochemical analysis in CRC patients identified substantial Hsp27 overexpression in CRC, highlighting its potential as a biomarker for early diagnosis, prognosis, and targeted therapy [17]. Moreover, elevated Hsp27 levels in the tumor stroma of CRC patients correlate with poorer clinical outcomes, particularly in those with lung metastases [45]. Additionally, Hsp27 overexpression has been observed in 5-FU-resistant human colon cancer WiDr-R cells, and siRNA-mediated knockdown of Hsp27 reduces this resistance [46].

Accumulating evidence from animal cancer models and in vivo studies further supports the critical role of Hsp27 in tumor progression. Elevated Hsp27 expression was linked to enhanced cancer cell proliferation, increased survival, and greater metastatic capacity. In lung cancer mouse models, increased Hsp27 drove rapid tumor growth by promoting cell proliferation, inhibiting apoptosis, and conferring resistance to anti-cancer drugs, resulting in more aggressive tumor phenotypes [47]. Similarly, upregulation of Hsp27 facilitates metastasis and invasiveness in prostate cancer models by modulating the epithelial–mesenchymal transition (EMT) [48].

Conversely, targeting Hsp27 with anti-sense oligonucleotides, shRNA, or specific inhibitors suppresses tumor growth by limiting proliferation, inducing apoptosis, and reducing metastasis. OGX-427, a well-characterized Hsp27 inhibitor, has demonstrated effectiveness in suppressing tumor growth and metastasis across several animal models of prostate, liver, and other cancers [49,50]. The miR-541-3p, a microRNA family member, specifically suppresses Hsp27 expression in prostate cancer cells [51]. Moreover, J2 compound, an Hsp27 inhibitor, reduced the proliferation of human ovarian cancer cell lines (SKOV3 and OVCAR-3) by inducing apoptotic pathways through its interaction with HSP27 [52].

Previous studies have shown that Hsp27 activity is correlated with activation of STAT3 and NF-κB transcription factors. Increased expression of Hsp27 modulated STAT3-mediated suppression of apoptosis in prostate cancer [53]. Overexpressed Hsp27 induced NF-kB activity, resulting in increased anti-apoptotic properties [54]. Overexpressed Hsp27 also reduced the cytochrome c release from mitochondria, thereby exerting anti-apoptotic effects through the inhibition of cytochrome c release [14]. We found that HK markedly increased total cytochrome c expression in CRC cells (Figure 4b) and significantly decreased Hsp27 protein levels (Figure 5). These findings suggest that elevated cytochrome c levels may be associated with inhibition of Hsp27 by HK treatment.

Up-regulated Hsp27 significantly increased the colony formation of HCC cells and invasion in normoxia and increased tolerance of hypoxia [55]. Knockdown of HSP27 in SW480 CRC cells significantly increased cell death, suppressed tumor growth, and enhanced sensitivity to the chemotherapeutic agents 5-fluorouracil and vincristine [22]. Furthermore, studies in breast cancer and melanoma models indicate that inhibiting Hsp27 not only slows tumor progression by reducing angiogenesis but also promotes long-term tumor dormancy [43,48]. Consistent with these observations, our findings showed that Hsp27 knockdown significantly inhibited the growth of CRC cells (Figure 8), and HK markedly reduced Hsp27 protein levels (Figure 5). These results suggest that modulation of Hsp27 may represent a promising therapeutic strategy for CRC, although further studies are needed to fully understand the underlying mechanisms. Collectively, these findings emphasize the multifaceted role of Hsp27 in cancer progression and therapeutic resistance, highlighting the importance of continued research into Hsp27-targeted therapies across diverse malignancies.

Recently, HK has been reported to specifically bind to the overexpressed oncogenic transcription factor FOXM1. This binding results in inhibition of FOXM1-mediated transcriptional activity and downregulation of FOXM1 protein expression [56]. It has also been shown to bind to overexpressed KRT18 in melanoma cells, leading to reduced KRT18 protein levels and consequent suppression of tumor growth both in vitro and in xenograft models [24]. HK has been shown through various experimental approaches to directly bind to the SLC3A2 protein in human monocyte THP-1 cells, a process that is associated with its anti-inflammatory effects [57]. It has been shown to bind to the unfolded ATPase domain of GRP78, thereby inducing the unfolded protein response (UPR) and triggering ER stress-mediated apoptosis [30]. Although several direct binding proteins of HK have been identified, detailed studies on the precise molecular mechanisms underlying HK binding to its target proteins remain limited.

Several studies utilizing molecular docking models have reported binding sites for HK target proteins. It has been shown to form hydrogen bonds with amino acid residues Lys721, Asp831, and Leu764 within the kinase domain of EGFR [58]. Additionally, docking studies have identified binding to key active site residues such as Val738 and Leu881 of RET, Leu825 of ErbB4, and Gln86 and His122 of Notch1 [59]. Furthermore, molecular docking studies have demonstrated interactions with residues His151, Arg152, and Lys243 at the active site of AMPK in glucosamine-treated HepG2 cells [60]. While these studies provide valuable insights into the amino acid residues potentially involved in these interactions, direct experimental evidence confirming the specific binding residues remains lacking.

Previous molecular docking and experimental research on HK target proteins has been limited, and no work to date has directly demonstrated a molecular interaction between HK and Hsp27. To our knowledge, this is the first study to provide strong biochemical and functional evidence that Hsp27 is a direct molecular target of HK in CRC cells. We showed that HK physically binds to Hsp27, which is particularly overexpressed in HEK 293T and CRC cells (Figure 6). Moreover, through molecular docking modeling, we have predicted the binding amino acid residues involved. By employing site-directed mutagenesis, we precisely identified Asn102 and His103 within the Hsp27 core domain as the critical residues that mediate this interaction (Figure 7). These findings uncover an original mechanism of action for HK’s anti-cancer effects and establish Hsp27 as a highly promising therapeutic target for direct intervention. Given the well-documented role of Hsp27 in CRC progression, metastasis, and drug resistance, our work provides new mechanistic insights and lays the groundwork for developing HK-based therapies targeting Hsp27.

Several studies have reported that HK affected various stress proteins, including heat shock proteins (HSPs). It inhibited HDAC6 activity, disrupting the HDAC6–Hsp90 interaction, which led to altered Hsp90 function and promoted its degradation in lung cancer cells [37]. Additionally, HK was shown to upregulate GRP78, an Hsp70 family molecular chaperone involved in ER stress that played a critical role in stress mitigation and cell survival [61]. Moreover, HK demonstrated cytoprotective effects by increasing heat shock protein 70 (Hsp70) expression in cadmium-induced liver injury in chickens [62]. Its treatment was reported to significantly reduce Hsp70 protein expression in multicentric chronic lymphocytic leukemia [63].

Despite these findings, few studies have elucidated the specific mechanisms or provided direct evidence of how HK modulates the expression or activity of Hsp27 or Hsp70 proteins. Previous research has mainly focused on the biological roles and pathways involving Hsp27 and Hsp70, while the direct regulatory effects of HK on these HSPs remain unclear and insufficiently demonstrated. In this study, we found that HK regulates HSPs in CRC cells, markedly reducing Hsp27 protein levels and selectively decreasing Hsp70 expression in DLD-1 cells (Figure 5). These results suggest that HK may exert anti-cancer effects across various cancers through its modulation of HSPs particularly Hsp27.

Previously, HK did not alter *KRT18* gene expression but decreased KRT18 protein levels in melanoma cells by promoting its degradation via ubiquitination [24]. On the one hand, HK upregulated the expression of UbcH8, an E2 ubiquitin-conjugating enzyme, thereby promoting the proteasomal degradation of the oncoprotein AML1-ETO in leukemia [64]. It has been reported that HK binds to FOXM1, inhibiting its transcriptional activity, which subsequently leads to a reduction in both FOXM1 mRNA and protein expression [56].

As demonstrated in Figure 5, our study further confirmed that HK treatment significantly decreased Hsp27 protein levels in CRC cells. However, whether this phenomenon reflects proteasomal degradation or suppressed expression remains unclear. Since studies on Hsp27 ubiquitination and degradation are limited, further research is needed to clarify the role of HK in regulating Hsp27 stability and chaperone activity.

A limitation of this study is the lack of in vivo validation for the anti-cancer effects of HK targeting Hsp27, which were observed in vitro. HK significantly inhibited CRC cell proliferation, induced apoptosis, and directly interacted with the Hsp27 under in vitro conditions. However, these findings may not fully reflect the complexities of the tumor microenvironment and in vivo factors that influence drug efficacy or modulate the HK–Hsp27 interaction. Therefore, further in vivo studies are warranted to confirm and better understand these effects.

Future studies will evaluate HK in well-established CRC animal models, such as xenografts and genetically engineered mice, to assess its therapeutic potential and target specificity for Hsp27 under physiological conditions. We also aim to investigate the sustained effects of HK on Hsp27 expression, related oncogenic pathways, and the physiological functions mediated by its direct binding to Hsp27. Together, these studies are expected to provide deeper insights into the mechanisms underlying HK’s anti-cancer effects and its potential synergy with standard treatments, thereby supporting further preclinical and clinical development.

## 5. Conclusions

This study successfully isolated and characterized HK from *M. officinalis* bark, confirming its purity by TLC, LC-MS/MS, and NMR, and quantified its content through HPLC for subsequent biological studies.

HK showed strong anti-cancer activity against CRC cells, significantly suppressing growth and colony formation in a dose-dependent manner. It also induced G0/G1 cell cycle arrest and apoptosis, evidenced by cytochrome c release, PARP cleavage, and reduced Bcl-2 expression. Thus, HK not only inhibits proliferation but also activates apoptotic signaling pathways.

A central finding was the identification of Hsp27 as HK’s direct molecular target. Pull-down and LC-MS/MS analyses revealed binding to Hsp27 core residues Asn102 and His103, confirmed by docking and mutational studies, suggesting HK acts as an inhibitor of Hsp27. Functionally, Hsp27 knockdown reduced CRC cell proliferation and anchorage-independent growth, while HK treatment markedly downregulated Hsp27 expression, with minor effect on Hsp70 and no effect on GRP78.

In summary, HK suppresses CRC cell growth in vitro, potentially involving inhibition of Hsp27, which leads cell cycle arrest and apoptosis. While our results suggest a role for Hsp27, further studies are required to clarify the extent of this dependency and to investigate additional molecular pathways contributing to HK’s anti-cancer effects. These findings support the therapeutic potential of HK as a natural anti-cancer agent and provide a rationale for further preclinical and clinical development.

## Figures and Tables

**Figure 1 cimb-47-00921-f001:**
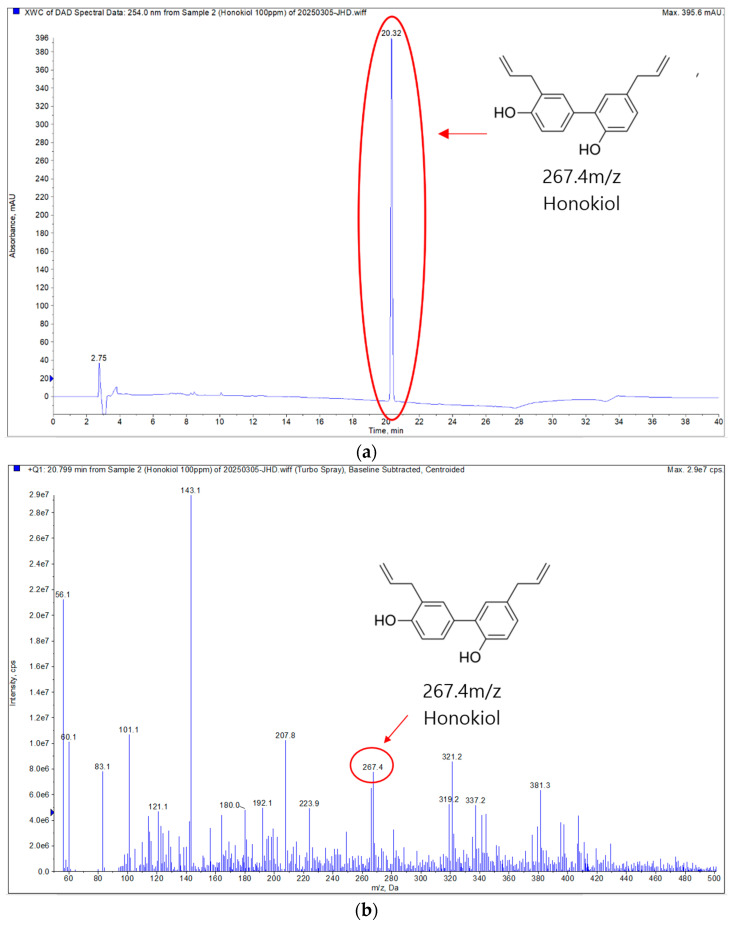
Positive ion mode LC-MS/MS analysis of the isolated compound (ODS Fr.1). (**a**) Extracted ion chromatogram (ECI) of ODS Fr.1. (**b**) Total ion chromatogram (TIC), with the highlighted region corresponding to the mass value of HK.

**Figure 2 cimb-47-00921-f002:**
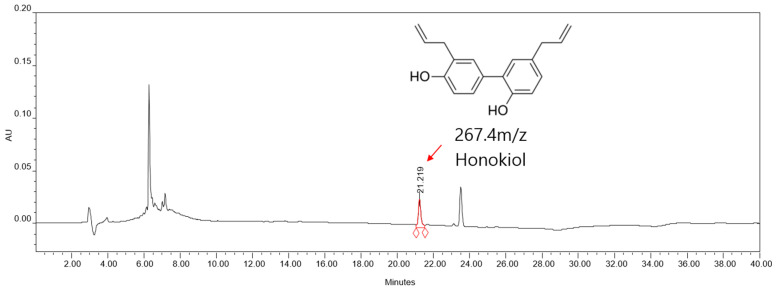
HPLC chromatogram of *M*. *officinalis* 60% EtOH extract at 1.0 mg/mL (1000 ppm).

**Figure 3 cimb-47-00921-f003:**
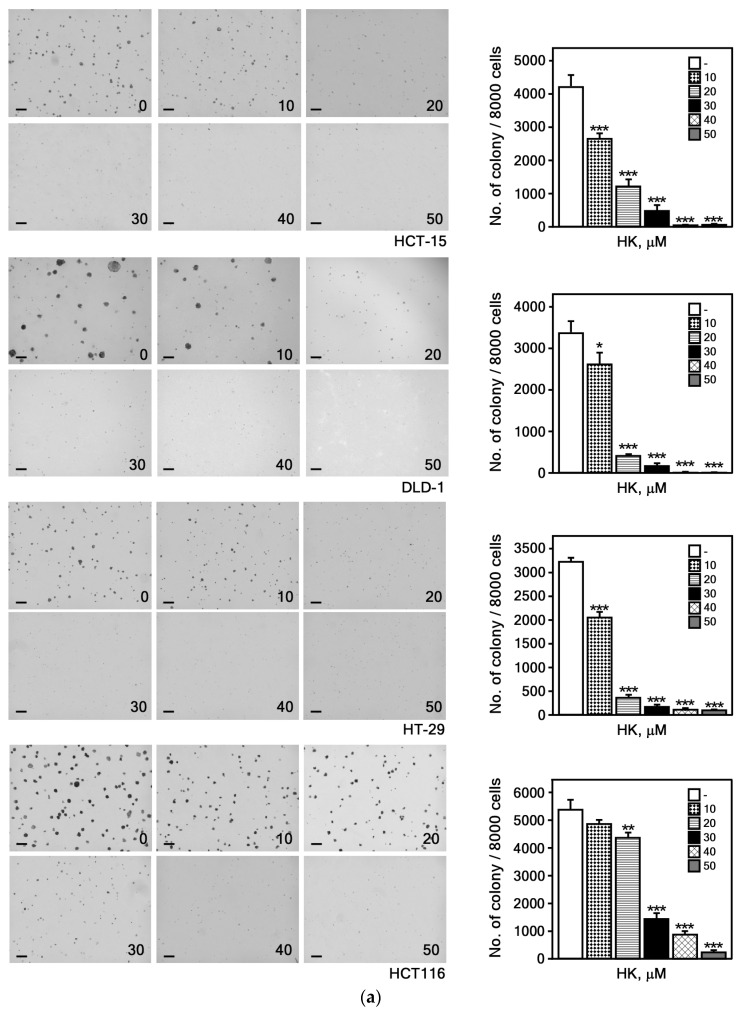
HK suppresses CRC cell growth and induces G0/G1 cell cycle arrest. (**a**) HK inhibits anchorage-independent colony growth of CRC cells in soft agar. Colonies were counted using a microscope and Image-Pro PLUS (v.6) software. Representative images (left) with a scale bar of 200 μm and quantification colony numbers (right) are presented as mean ± S.E.M. (*n* = 3). Differences were considered significant based on statistical analysis. (**b**) HK suppresses CRC cell proliferation in a dose-dependent manner. Cells were treated with HK (0, 10, 20, 30, 40, or 50 μM), and cell growth was measured at 48 or 72 h after treatment. Values are expressed as mean ± S.E.M (*n* = 8). Statistical significance was determined using ANOVA (*, *p* < 0.05; **, *p* < 0.01; ***, *p* < 0.001, compared with untreated controls). (**c**) HK induces G0/G1 cell cycle arrest in CRC cells. Cells were treated with HK (0 or 30 μM), and cell cycle distribution was analyzed by flow cytometry. Data are presented as mean ± S.D. (*n* = 3). Student’s *t*-test was used for comparison (*, *p* < 0.01; **, *p* < 0.001; ***, *p* < 0.0001, compared with untreated controls).

**Figure 4 cimb-47-00921-f004:**
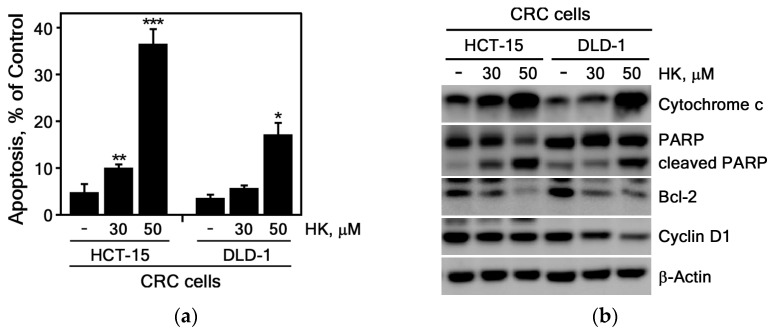
HK induces apoptosis and alters the expression of apoptosis-associated proteins in CRC cells. (**a**) HK induces apoptosis in CRC cells. HCT-15 and DLD-1 cells were treated with HK (0, 30, or 50 μM) and analyzed for apoptosis by flow cytometry after staining with annexin V-FITC and propidium iodide (PI). Apoptosis is expressed as a percentage relative to untreated controls. Data represents the mean ± S.D. (*n* = 3). Statistical significance was determined using Student’s *t*-test (*, *p* < 0.01; **, *p* < 0.001; ***, *p* < 0.0001 compared with untreated controls). (**b**) HK modulates apoptosis-related proteins in CRC cells. HCT-15 and DLD-1 cells were treated with HK (0, 30, or 50 μM), and the expression levels of cytochrome c, Bcl-2, cleaved PARP, and cyclin D1 were examined by Western blotting. β-Actin served as a loading control.

**Figure 5 cimb-47-00921-f005:**
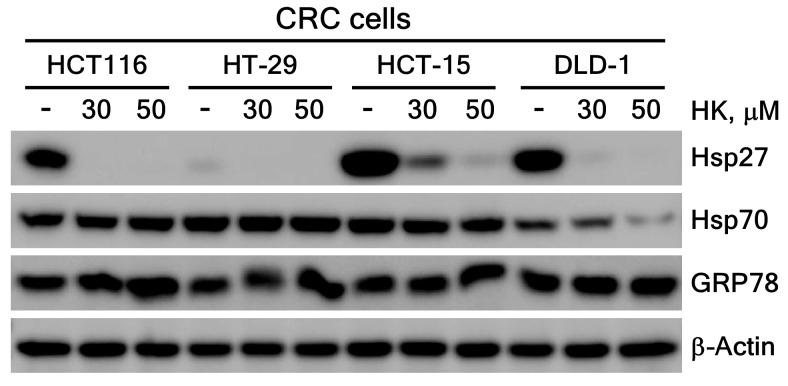
Expression levels of Hsp27, Hsp70, and GRP78 proteins following HK treatment in CRC cells. Hsp27, Hsp70, and GRP78 were detected by immunoblotting, with β-actin used as a loading control.

**Figure 6 cimb-47-00921-f006:**
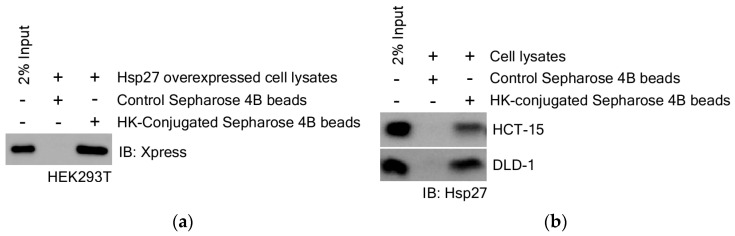
HK directly binds to Hsp27. (**a**) Direct binding of HK to overexpressed Hsp27 in HEK293T cell lysates. (**b**) Binding of HK to endogenous Hsp27 in HCT-15 and DLD-1 CRC cell lysates. Cell lysates were incubated with control Sepharose 4B beads or HK-conjugated Sepharose 4B beads, and bound proteins were detected by immunoblotting.

**Figure 7 cimb-47-00921-f007:**
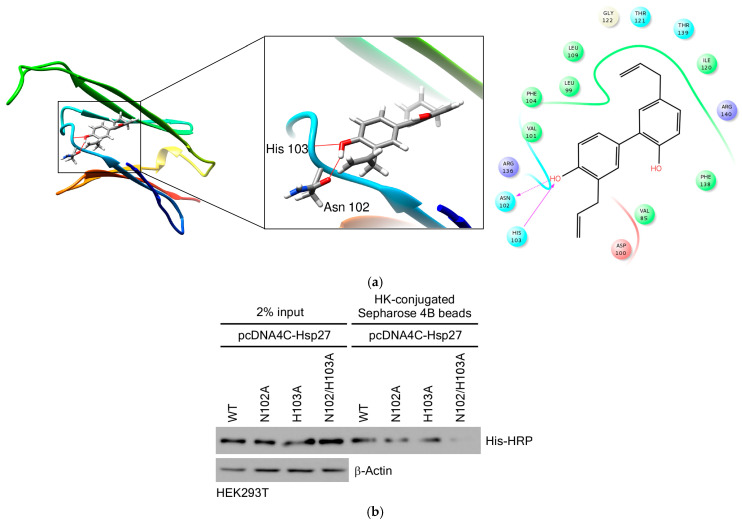
HK interacts with the core region of Hsp27. (**a**) Molecular modeling of the HK-Hsp27 interaction. Left, enlarged view of the interaction interface; Right, Ligand Interaction Diagram (LID). Hsp27 is shown in ribbon representation and HK as a stick model. Hydrogen bonds (red lines) were observed between HK and the Asn102/His103 residues in the Hsp27 core region. (**b**) Binding of HK to wild-type, single, and double mutant Hsp27 proteins. Wild-type, single, and double mutant Hsp27 were expressed in HEK293T cells using *pcDNA4C-Hsp27*, and lysates were incubated with either control or HK-conjugated Sepharose 4B beads. Bound proteins were analyzed by immunoblotting to assess the binding efficiencies of wild-type and mutant Hsp27 for HK. HK bound to wild-type, single, and double mutant Hsp27 was detected with an 6X His-HRP antibody, and β-actin was used as a loading control.

**Figure 8 cimb-47-00921-f008:**
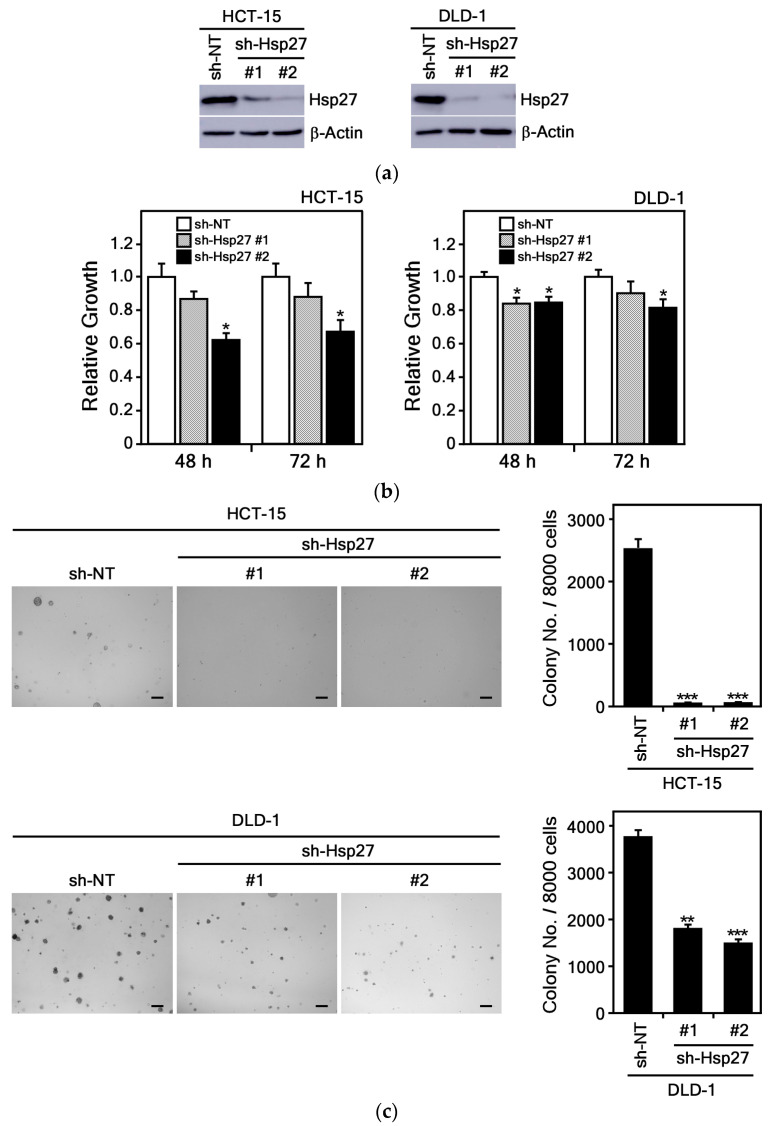
Hsp27 knockdown suppresses proliferation and colony formation in CRC cells. (**a**) Endogenous Hsp27 protein expression was examined by immunoblotting in HCT-15 and DLD-1 cells stably infected with *sh-NT* or *sh-Hsp27*. (**b**) Hsp27 knockdown reduced proliferation of HCT-15 and DLD-1 CRC cells, measured by absorbance at 492 nm at 48 and 72 h. Data represents the mean ± S.D. (*n* = 3). (**c**) Anchorage-independent growth was decreased by Hsp27 knockdown in HCT-15 and DLD-1 CRC cells. Representative images (left) with a scale bar of 200 μm and quantification colony numbers (right) are presented as mean ± S.D. (*n* = 8). Statistical significance was determined by Student’s *t*-test; asterisks indicate decreases in *sh-Hsp27* CRC cells compared with *sh-NT* control (*, *p* < 0.05; **, *p* < 0.01; ***, *p* < 0.001 compared with untreated controls).

**Table 1 cimb-47-00921-t001:** Full hairpin sequences of *sh-NT* and *sh-Hsp27*.

Name	Full Hairpin Sequence (5′→3′)
*Sh-NT*	ccgggcaagctgaccctgaagttcactcgagtgaacttcagggtcagcttgcttttt
*Sh-Hsp27* #1	ccggcagtccaacgagatcaccatcctcgaggatggtgatctcgttggactgtttttg
*Sh-Hsp27* #2	ccgggatcaccatcccagtcaccttctcgagaaggtgactgggatggtgatctttttg

**Table 2 cimb-47-00921-t002:** Sequences of specific point-mutated primers used for Hsp27 site-directed mutagenesis.

Primer Name	Sequence of Primers (5′→3′)	Description
N102A-FP	cctggatgtcgcccacttcgccccggacgagc	Forward primer
N102A-RP	gctcgtccggggcgaagtgggcgacatccagg	Reverse primer
H103A-FP	ggatgtcaacgccttcgccccggacgagctga	Forward primer
H103A-RP	tcagctcgtccggggcgaaggcgttgacatcc	Reverse primer
N102/H103A-FP	tccctggatgtcgccgccttcgccccggacgagctgaac	Forward primer
N102/H103A-FP	gttcagctcgtccggggcgaaggcggcgacatccaggga	Reverse primer

**Table 3 cimb-47-00921-t003:** Identification of peptides binding to HK-conjugated Sepharose 4B beads in CRC cell lysates.

Identified Proteins	Peptides (95%)
Hsp27	1
Hsp70	1
NUP93	1
KRT9	1

## Data Availability

The original contributions presented in this study are included in the article/Appendix A. Further inquiries can be directed to the corresponding authors.

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
