# Peer review of "Honokiol Inhibits Colorectal Cancer Cell Growth: Involvement of Hsp27 as a Molecular Target"

_cimb, 2025, doi:10.3390/cimb47110921_

Round 1
Reviewer 1 Report
Comments and Suggestions for Authors
I have read and considered manuscript cimb-3912539 – “Honokiol inhibits colorectal cancer cells growth by targeting Hsp27 protein and downregulating its oncogenic activity” by Kim et al.
Honokiol has long been used in Chinese medicine and scientists have been studying both its anti-cancer activity and mechanism of action for almost 50 years. Papers early in the 2000s implicate that its mechanism seems to involve a G0/G1 block and a link between honokiol and heat shock proteins was established 7 or 8 years ago. In this manuscript the authors indicate that they provide evidence for a direct interaction between HSP27 and honokiol and that this interaction is part of the reason honokiol is an anti-cancer agent.
Unfortunately, the data they present, in this manuscript, is not compelling for the mechanism
the authors are suggesting.
It is not clear why the extraction and purification of honokiol needs to be described in detail in this manuscript. Why not use synthetic honokiol?
Down regulation of HSP27 does inhibit the proliferation of some CRC cell lines, but others are less sensitive. Their initial data shows that HSP27 inhibits anchorage independent proliferation of CRC cell lines, but honokiol is not very potent (IC50 >10 micromolar).
It was not clear why data in both figure 3a and 3b are included – they seem to show much the same information. It might be interesting to know the effects at 96hrs or even 120 hours, but the authors do not discuss their 72 hour data.
The evidence presented for an interaction between honokiol and hsp27 is totally in adequate. HSP proteins are often non-specific contaminants of affinity pull-downs. The specificity of the pulldowns is not addressed adequately. The yields of hsp27 protein, the Coomassie blue stained pull-downs are not shown. The complexity of the protein pulldowns is not addressed adequately. The molecular modelling assumes the interaction is significant, it does not provide evidence for the interaction. The gel showing the effects of the mutations on the pulldowns (fig 5) sems to have been mislabelled.
Knockdown of HSP27 in DLD1 has only a small effect on the clonogenicity of these cells in soft-agar. Honokiol seems to have reduced the levels of HSP70 in DLD-1 cells (Figure 8).How would this alter the interpretation of these experiments? These experiments do not make a strong case for the specific involvement of HSP27 in the mechanism of action of the anti-cancer activity of honokiol.
As described the experiments do not inform us about the mechanism of action of honokiol.
Author Response
Reviewer 1
Dear Reviewer
We are sincerely thankful for reviewer’s thorough evaluation and constructive feedback, which have greatly contributed to improving our manuscript. We acknowledge and respect the concerns raised regarding the mechanistic insights into honokiol’s anti-cancer activity and the specificity of its interaction with HSP27. While our study provides meaningful evidence supporting these interactions and their biological relevance, we agree that further detailed investigations are warranted. We have carefully addressed the reviewers’ comments and revised the manuscript accordingly to clarify the scope and interpretation of our findings. We respectfully ask for the reviewers’ and editors’ kind reconsideration of our revised submission in light of these improvements.
Comment 1:
Honokiol has long been used in Chinese medicine and scientists have been studying both its anti-cancer activity and mechanism of action for almost 50 years. Papers early in the 2000s implicate that its mechanism seems to involve a G0/G1 block and a link between honokiol and heat shock proteins was established 7 or 8 years ago. In this manuscript the authors indicate that they provide evidence for a direct interaction between HSP27 and honokiol and that this interaction is part of the reason honokiol is an anti-cancer agent.
Unfortunately, the data they present, in this manuscript, is not compelling for the mechanism the authors are suggesting.
Response 1:
We sincerely appreciate the reviewer’s thoughtful evaluation and constructive comments on our manuscript. As with many natural compounds, honokiol has been reported to exert diverse beneficial effects. As the reviewer correctly noted, previous studies have shown that honokiol induces G0/G1 cell cycle arrest and decreases the expression of heat shock proteins. In recent years, particular attention has been given to identifying specific amino acid residues involved in direct interactions and binding within the core regions of such proteins. Moreover, many ongoing studies are focusing on the active sites of target proteins and their critical amino acid residues.
As the reviewer also pointed out, our present study does not provide direct mechanistic evidence for honokiol–Hsp27 interaction leading to protein degradation and inhibition, which indeed requires further investigation. Nevertheless, we were able to demonstrate that honokiol binds to a specific amino acid residue within the core region of Hsp27. In addition, we observed that honokiol treatment significantly reduced Hsp27 protein levels in colon cancer cells, without comparable effects on other heat shock proteins. We believe these findings provide indirect yet meaningful evidence for honokiol’s selective interaction with Hsp27.
We fully agree with the reviewer that elucidating the downstream molecular events following the honokiol–Hsp27 interaction would allow us to establish a more explicit and convincing mechanism. We sincerely hope the reviewer will take this into consideration when evaluating the current study.
Comment 2:
It is not clear why the extraction and purification of honokiol needs to be described in detail in this manuscript. Why not use synthetic honokiol?
Response 2:
Our laboratory specializes in isolating and purifying natural single compounds and has published several studies in CIMB and Pharmaceuticals. As the reviewer mentioned, initial studies utilized commercially available honokiol to lay the foundation of this research. However, subsequent in vitro and animal experiments required larger amounts of honokiol, which led us to employ our specialized purification techniques. A detailed description of these methods, along with their advantages, can be found in the first paragraph of Section 4, Discussion. We consider including this information important to ensure the quality and reproducibility of the honokiol used in our experiments.
Comment 3:
Down regulation of HSP27 does inhibit the proliferation of some CRC cell lines, but others are less sensitive. Their initial data shows that HSP27 inhibits anchorage independent proliferation of CRC cell lines, but honokiol is not very potent (IC50 >10 micromolar).
Response 3:
We sincerely appreciate the reviewer’s thoughtful observations regarding the specific involvement of HSP27 in the mechanism of honokiol’s anti-cancer activity. In the soft-agar assay results, we note that knockdown of HSP27 in DLD-1 cells leads to a partial but incomplete reduction in clonogenicity, suggesting that HSP27 may contribute to, but is not solely responsible for, the anchorage-independent growth of these cells. This partial effect could reflect functional redundancy or compensation by other heat shock proteins within the cell, which remain unaffected by HSP27 knockdown.
Furthermore, as shown in Figure 5 of revised manuscript, treatment with honokiol not only reduces the levels of HSP27 but also significantly lowers HSP70 expression in DLD-1 cells. This broader impact on multiple heat shock proteins suggests that honokiol’s anti-cancer mechanism may involve inhibition of a network of chaperones rather than exclusive targeting of HSP27. The concomitant downregulation of HSP70 in DLD-1 cells indicates that honokiol could exert its effects through additional pathways, thus attenuating the overall protective chaperone environment in colorectal cancer cells. We also updated Figure 8 to reflect Figure 5 in the revised manuscript, and provided supplementary information regarding this.
Taken together, our findings indicate that while HSP27 is functionally relevant, it may not be the only mediator of honokiol’s anti-cancer activity in DLD-1 cells. We acknowledge that these experiments do not provide definitive evidence for a specific and exclusive role of HSP27. As such, further investigation is warranted to fully delineate the interplay and compensatory mechanisms among heat shock proteins in the context of honokiol treatment. We thank the reviewer again for highlighting these important considerations, which have helped refine our interpretation and future direction of the study.
Comment 4:
It was not clear why data in both figure 3a and 3b are included – they seem to show much the same information. It might be interesting to know the effects at 96hrs or even 120 hours, but the authors do not discuss their 72 hour data.
Response 4:
We sincerely appreciate the reviewer’s insightful suggestion regarding Figures 3a and 3b. While both figures indeed demonstrate the inhibitory effects of honokiol on the growth of colon cancer cells, they represent distinct biological aspects. Specifically, Figure 3a shows the results of the soft-agar assay, which assesses anchorage-independent growth—an important hallmark of tumorigenic potential. In contrast, Figure 3b presents cell viability data, reflecting the survival of colon cancer cells. Therefore, together these figures provide complementary evidence that honokiol suppresses both growth and tumorigenicity of colorectal cancer cells. We have clarified this point in the first paragraph of Section 3.2.1. (Line 348-354)
Regarding the time-course data, we agree with the reviewer that it would be interesting to assess effects at 96 or 120 hours. However, under our optimized experimental conditions, we observed that cell growth rates after 48 and 72 hours provide the most reliable and clear indication of honokiol’s effect. Beyond 72 hours, both treated and untreated cells exhibited decreased growth and increased cell death, making interpretation of longer time points more challenging. We have added an explanation of these considerations in the revised manuscript.
We thank the reviewer again for raising these important points which have helped improve the clarity and rigor of our study.
Comment 5:
The evidence presented for an interaction between honokiol and hsp27 is totally in adequate. HSP proteins are often non-specific contaminants of affinity pull-downs. The specificity of the pulldowns is not addressed adequately. The yields of hsp27 protein, the Coomassie blue stained pull-downs are not shown. The complexity of the protein pulldowns is not addressed adequately. The molecular modelling assumes the interaction is significant, it does not provide evidence for the interaction. The gel showing the effects of the mutations on the pulldowns (fig 5) sems to have been mislabelled.
Response 5:
We agree with the reviewer’s concern that HSP proteins can act as nonspecific contaminants in affinity pulldown experiments. In addition, we recognize that the abundance of honokiol target proteins and potential nonspecificity of antibodies are important factors to consider. In this study, we confirmed the binding of honokiol to Hsp27 in cells overexpressing a specifically tagged Hsp27 protein, as shown in Figure 6a and Table 1 via LC-MS/MS analysis. Furthermore, binding of honokiol to endogenous Hsp27 overexpressed in colon cancer cells was also verified. These results collectively provide evidence supporting the direct interaction between honokiol and Hsp27.
Regarding the reviewer’s suggestion about assessing Hsp27 protein yield by Coomassie blue-stained pulldown using purified Hsp27, we note that our experiment identified honokiol-bound Hsp27 in total cell lysates by Western blot following pulldown, which does not allow for precise quantitative analysis. Nevertheless, the method employed is well-established and widely used by researchers.
Computational modeling remains a valuable tool for predicting the amino acid residues involved in honokiol-Hsp27 interaction and the binding affinity, and is commonly used in this research field. In our study, computer simulations indicated possible hydrogen bonding between honokiol and Asn102 and His103 in the Hsp27 core region. This hypothesis was further supported by site-directed mutagenesis substituting these residues with alanine, where pull-down assays demonstrated significantly reduced honokiol binding to mutant compared to wild-type Hsp27 (Figure 7b, revised text).
However, direct evidence that honokiol binding affects Hsp27 function is still lacking. We did observe a significant decrease in Hsp27 activity upon honokiol treatment, but whether this results from altered gene expression or protein degradation caused by binding requires further investigation. As the reviewer suggested, future studies should quantify honokiol-Hsp27 binding and elucidate the functional consequences of this interaction.
Comment 6:
As described the experiments do not inform us about the mechanism of action of honokiol.
Response 6:
We sincerely appreciate the reviewer’s thoughtful comments and concerns, and kindly request your careful consideration of the study presented herein.
Reviewer 2 Report
Comments and Suggestions for Authors
The manuscript investigates the anticancer effects of honokiol (HK) in colorectal cancer (CRC) cells, focusing on its ability to inhibit proliferation, induce apoptosis, and directly interact with heat shock protein 27 (Hsp27). The authors combine cell-based assays, apoptosis markers, molecular docking, mutagenesis, and shRNA knockdown to support their conclusions. The work is comprehensive and provides novel insights into HK as a potential therapeutic candidate targeting Hsp27.
While the study is methodologically solid, several aspects of the experimental design, data interpretation, and presentation require clarification or additional work before publication.
Major Comments
- Although HK binding to Hsp27 is demonstrated, it remains unclear whether HK’s anticancer effects are dependent on Hsp27. This could be done by testing the HK effects in Hsp27-knockdown cells. If HK activity is reduced, this would directly establish causality.
- Docking results are predictive and without molecular dynamics (MD) simulations or binding affinity quantification the conclusions about “specific binding” remain preliminary.
- The stated hypothesis suggesting that HK directly binds to Hsp27 and attenuates its anti-apoptotic function—appears to be informed by experimental outcomes rather than being an a priori, testable prediction. The current phrasing gives the impression of a post hoc interpretation rather than a hypothesis that guided the study from the outset. I recommend revising the Introduction to clearly distinguish between the study’s initial hypothesis (based on existing literature or preliminary observations) and the novel insights gained through the experimental work. Table 1. Identification of peptides binding to HK-conjugated Sepharose 4B beads in CRC cell lysates – there are identified proteins, but only the first (Hsp27) is tested. Give an explanation why in the text.
- Statistical Analysis - many figures involve multiple groups, yet the manuscript consistently uses Student’s t-test. ANOVA with appropriate post-hoc tests would be more rigorous.
Minor Comments
- Introduction is too long and could be more concise because there are too many details on individual studies – some of the references could be mentioned in the discussion. Try focusing on CRC studies when mentioning HK and Hsp27 studies.
- Some functional assays are performed on CRC cell lines, and some only on two. Insert the explanation in the manuscript.
- Apoptosis markers described as Figure 5b, likely meant Figure 4b.
- Please add more detail in flow cytometry Material and methods section (e.g. machine, software used)
- Figure 3 “Data are presented as mean ± S.D.” is repeated
- Line 80 - Furthermore, Moreover is written – double check
- lines 457-460 similar to text in lines 470-476
Author Response
Reviewer 2
Dear Reviewer
We sincerely thank you for your thorough and constructive feedback on our manuscript. Your valuable suggestions have helped us improve the clarity and quality of our work.
We have addressed your major comments by clarifying HK’s dependence on Hsp27, acknowledging the limits of docking data while providing complementary experimental evidence, and revising the Introduction to distinguish between our initial hypothesis and novel findings. The rationale for focusing on Hsp27, statistical analyses using ANOVA, and flow cytometry details were also clarified and expanded as recommended. Minor comments regarding introduction length, cell line selection, figure citations, repetition, and editorial issues were carefully revised to enhance clarity and accuracy.
We hope our revisions meet your expectations and look forward to your continued consideration.
<Major Comments>
Comment 1:
Although HK binding to Hsp27 is demonstrated, it remains unclear whether HK’s anticancer effects are dependent on Hsp27. This could be done by testing the HK effects in Hsp27-knockdown cells. If HK activity is reduced, this would directly establish causality.
Response 1:
We sincerely appreciate the reviewer’s thoughtful consideration. We understand the reviewer’s concern regarding the unclear dependence of HK’s anticancer effects on Hsp27. Establishing a causal relationship between HK activity and reduced Hsp27 expression in Hsp27-knockdown colon cancer cell lines would indeed provide valuable evidence supporting anticancer strategies targeting Hsp27. In our study, Hsp27 knockdown resulted in a significant reduction in colon cancer cell growth at the cellular level compared to HK treatment alone. This finding suggests that targeting Hsp27 may offer an alternative therapeutic approach for cancers with Hsp27 overexpression. We agree that demonstrating this causal relationship in animal models or clinical settings would further strengthen its significance as a cancer treatment strategy. We therefore consider that future studies using appropriate animal models and clinical samples will be essential to clarify this issue more definitively.
Comment 2:
Docking results are predictive and without molecular dynamics (MD) simulations or binding affinity quantification the conclusions about “specific binding” remain preliminary.
Response 2:
The authors fully agree with the reviewer’s insightful comment regarding the need for molecular dynamics (MD) simulations and binding affinity quantification to strengthen conclusions about “specific binding.” Such studies require purified wild-type and mutant Hsp27 proteins, and we believe these future investigations will provide more definitive and detailed evidence. Pull-down assays targeting specific proteins are also widely recognized as valuable methods for confirming direct interactions, and many researchers employ these techniques. In the current study, we confirmed the binding of HK to endogenous Hsp27. Additionally, by overexpressing wild-type and mutant Hsp27 proteins tagged with specific markers in cell-based systems, we demonstrated the binding affinity of HK to the amino acid residues predicted by our computational modeling. We believe that this combined evidence of specific binding provides a solid foundation for further studies and potential clinical applications targeting HK-overexpressed Hsp27. We kindly ask the reviewer to consider these points.
Comment 3:
The stated hypothesis suggesting that HK directly binds to Hsp27 and attenuates its anti-apoptotic function—appears to be informed by experimental outcomes rather than being an a priori, testable prediction. The current phrasing gives the impression of a post hoc interpretation rather than a hypothesis that guided the study from the outset. I recommend revising the Introduction to clearly distinguish between the study’s initial hypothesis (based on existing literature or preliminary observations) and the novel insights gained through the experimental work.
Response 3:
Thank you for your valuable comment. In response to your suggestion, we have revised the final paragraph of the Introduction to clearly distinguish the study’s initial hypothesis—based on existing literature and preliminary observations—from the novel insights gained through our experimental work. This modification aims to clarify that the hypothesis guided the study from the outset, while the subsequent findings provide new mechanistic understanding. We believe this revision improves the clarity and logical flow of the manuscript.
Comment 4:
Table 1. Identification of peptides binding to HK-conjugated Sepharose 4B beads in CRC cell lysates – there are identified proteins, but only the first (Hsp27) is tested. Give an explanation why in the text.
Response 4:
The reason for validating only Hsp27, as proposed by the author, is explained in the first paragraphs of Sections 3.3.2 and 3.3.3 of the main text. In addition, to further support the rationale for focusing on Hsp27, Figure 8 was revised to show the results of experiments on four types of CRC cells and was moved to Figure 5. Moreover, unpublished experimental results confirmed that HK binds to Hsp70; however, Hsp27 was selected because its protein levels were markedly reduced after HK treatment in colon cancer cells.
Comment 5:
Statistical Analysis - many figures involve multiple groups, yet the manuscript consistently uses Student’s t-test. ANOVA with appropriate post-hoc tests would be more rigorous.
Response 5:
We fully agree with the statistical analysis approach suggested by the reviewer. Accordingly, we applied ANOVA to analyze the anchorage-independent assay and MTS assay experiments involving three or more groups in HK-treated CRC cells. These analyses are detailed in Section 2.14 and presented in Figures 3a and 3b.
<Minor Comments>
Comment 1:
Introduction is too long and could be more concise because there are too many details on individual studies – some of the references could be mentioned in the discussion. Try focusing on CRC studies when mentioning HK and Hsp27 studies.
Response 1:
We appreciate the reviewer’s suggestion. In response to the reviewer’s comments, the introduction has been substantially revised, with redundant content removed, to focus more directly on CRC, honokiol, and Hsp27 (see Introduction). In addition, some of the content discussed in the Introduction has been moved to the Discussion section.
Comment 2:
Some functional assays are performed on CRC cell lines, and some only on two. Insert the explanation in the manuscript.
Response 2:
At the reviewer's suggestion, the reason for using two cell lines in the apoptosis assay was explained in lines 411-414 of section 3.2.2 of the manuscript.
Comment 3:
Apoptosis markers described as Figure 5b, likely meant Figure 4b.
Response 3:
In accordance with the reviewer’s suggestion, we have corrected line 37-38 of Section 3.2.2 in the manuscript by changing 'Figure 5b' to 'Figure 4b.' We sincerely appreciate the reviewer’s careful consideration.
Comment 4:
Please add more detail in flow cytometry Material and methods section (e.g. machine, software used)
Response 4:
As suggested by the reviewer, details of the instrument and software used for flow cytometry analysis have been included in Sections 2.11 and 2.12.
Comment 5:
Figure 3 “Data are presented as mean ± S.D.” is repeated
Response 5:
In response to the reviewer’s suggestion, we have removed the repeated sentence “Data are presented as mean ± SD” in Figure 3. This was our oversight, and we sincerely appreciate the reviewer’s careful observation.
Comment 6:
Line 80 - Furthermore, Moreover is written – double check
Response 6:
We thank the reviewer for pointing out the duplication on Line 80. In accordance with the reviewer’s suggestion, we have fixed the duplicated expressions in the introduction during revision. We apologize for this oversight and sincerely appreciate the reviewer’s careful attention to detail.
Comment 7:
Lines 457-460 similar to text in lines 470-476
Response 7:
We appreciate the reviewer’s thoughtful observation regarding the overlap between lines 457-460 and lines 470-476 of original manuscript. In response, we have carefully revised the manuscript at line 452-454 to address these redundancies. In particular, the last paragraph of Section 3.3.3. of revised manuscript has been deleted to eliminate duplication and improve clarity.
Round 2
Reviewer 1 Report
Comments and Suggestions for Authors
The authors' explanations are not convincing.
The manuscript still has significant flaws and
should not be published as is.
Author Response
Reviewer 1
Dear Reviewer
We deeply appreciate the reviewer’s careful evaluation and valuable suggestions regarding this study.
Recent research on the anticancer effects of small molecules, including natural products, has primarily focused on elucidating their molecular mechanisms and anticancer efficacy through molecular dynamics and docking models. For example, in the study "Discovery of Novel HSP27 Inhibitors as Prospective Anti-Cancer Agents Utilizing Computer-Assisted Therapeutic Discovery Approaches" by Umar et al. (2022) (https://pmc.ncbi.nlm.nih.gov/articles/PMC9368632/), the involvement of Hsp27 in cancer treatment resistance was investigated, and novel inhibitors were identified through docking models. Additionally, Iwuchukwu & Achilonu (2025) demonstrated in their study, "Targeting Hsp27 inhibition in Glioblastoma: A comprehensive in silico investigation" (https://www.sciencedirect.com/science/article/pii/S1093326325001925), that the Arg140 and His103 amino acid residues of Hsp27 are key to its stability. Furthermore, Nappi et al. (2020) reported in "Ivermectin inhibits HSP27 and potentiates the efficacy of oncogene targeting in tumor models" (https://www.jci.org/articles/view/130819) that ivermectin selectively binds to Hsp27 and inhibits protein aggregation without affecting Hsp70 and Hsp90. Liu et al. (2020), in their study "Honokiol Inhibits Melanoma Growth by Targeting Keratin 18 In Vitro and In Vivo" (https://pubmed.ncbi.nlm.nih.gov/33330500/), demonstrated that honokiol binds to Keratin 18 and promotes its ubiquitination-mediated degradation, thereby inhibiting melanoma growth. Numerous other studies have explored the anticancer effects of natural products, including honokiol, and their molecular target proteins.
In our study, we also elucidated the interaction and binding sites between HK and Hsp27 through computational docking modeling. Although the clinical relevance of the HK-Hsp27 interaction and the specific functional changes in Hsp27 induced by this interaction have not yet been fully clarified, we plan to address these aspects in future research. We kindly ask the reviewers to consider this context when evaluating the revised manuscript.
Thank you for your time and thoughtful consideration.
Reviewer 2 Report
Comments and Suggestions for Authors
I suggest to the authors to soften the conclusion of HK’s anticancer effects on dependence of Hsp27, since I do not see any changes have been made in the manuscript regarding new experiments.
Rest of my comments has been addressed.
Author Response
Reviewer 2
Dear Reviewer
We would like to express our sincere gratitude for your thorough review and thoughtful suggestions regarding our manuscript. In accordance with your valuable feedback, we have carefully revised the title and relevant sections to better reflect your comments. We respectfully ask you to review the revised version, taking into account the modifications that have been made.
Comment 1:
I suggest to the authors to soften the conclusion of HK’s anticancer effects on dependence of Hsp27, since I do not see any changes have been made in the manuscript regarding new experiments.
Rest of my comments has been addressed.
Response 1:
We appreciate the reviewer’s thoughtful evaluation of our study and valuable suggestion. In response to the recommendation to soften the conclusion regarding the dependence of HK’s anticancer effects on Hsp27, we have revised the manuscript accordingly. Specifically, we modified the title from “Honokiol inhibits colorectal cancer cell growth by targeting Hsp27 protein and downregulating its oncogenic activity” to “Honokiol inhibits colorectal cancer cell growth: involvement of Hsp27 as a molecular target”. In addition, relevant sections have been revised to present this relationship in a more cautious manner. These changes can be found in: 1) Abstract (lines 35–38), 2) Introduction (lines 97–101), 3) Results (lines 463–466), 4) Discussion (lines 585–590, 661–662, 665–666), and 5) Conclusions (lines 686–690).